# The Healthy Environments and Active Living for Translational Health (HEALTH) Platform: A smartphone-based system for geographic ecological momentary assessment research

Alexander J. Wray[1,2,¤a,*], Katelyn R. O'Bright[3,¤b], Shiran Zhong[1,2], Sean Doherty[2,3], Michael Luubert[4], Jed Long[1,2], Catherine E. Reining[1,2,3], Christopher J. Lemieux[3], Jon Salter[4], Jason Gilliland[1,2,5]

1 Department of Geography and Environment, Western University, London, Ontario, Canada, 2 Human Environments Analysis Lab, Western University, London, Ontario, Canada, 3 Department of Geography and Environmental Studies, Wilfrid Laurier University, Waterloo, Ontario, Canada, 4 Education & Research, Esri Canada Limited, Toronto, Ontario, Canada, 5 Department of Epidemiology & Biostatistics, Department of Paediatrics, School of Health Studies, Western University, London, Ontario, Canada

¤a Current address: Department of Public Health, University of Tennessee Knoxville, Knoxville, Tennessee, United States of America
¤b Current address: Cultural Geography Group, Wageningen University & Research, Wageningen, Gelderland, The Netherlands
* awray22@uwo.ca

## Abstract

Smartphones have become a widely used tool for delivering digital health interventions and conducting observational research. Many digital health studies adopt an ecological momentary assessment (EMA) methodology, which can be enhanced by collecting participant location data using built-in smartphone technologies. However, there is currently a lack of customizable software capable of supporting geographically explicit research in EMA. To address this gap, we developed the Healthy Environments and Active Living for Translational Health (HEALTH) Platform. The HEALTH Platform is a customizable smartphone application that enables researchers to deliver geographic ecological momentary assessment (GEMA) prompts on a smartphone in real-time based on spatially complex geofence boundaries, to collect audiovisual data, and to flexibly adjust system logic without requiring time-consuming updates to participants' devices. We illustrate the HEALTH Platform's capabilities through a study of park exposure and well-being. This study illustrates how the HEALTH Platform improves upon existing GEMA software platforms by offering greater customization and real-time flexibility in data collection and prompting participants. We observed survey prompt adherence is associated with participant motivation and the complexity of the survey instrument itself, following past EMA research findings. Overall, the HEALTH Platform offers a flexible solution for implementing GEMA in digital health research and practice.

**Data availability statement:** Data is subject to access limitations due to confidentiality and ethical research requirements. Please contact the researchers via heal@uwo.ca for further inquiries.

**Funding:** This research was supported by funding from the Public Health Agency of Canada to JG. The views expressed herein do not necessarily represent the view of the Public Health Agency of Canada. The funder had no role in study design, data collection and analysis, decision to publish, or preparation of the manuscript.

**Competing interests:** The authors have declared that no competing interests exist.

## Author summary

Smartphones are a powerful tool for observing and changing health-related behaviours. Ecological momentary assessment is the research method that is commonly used to undertake this kind of observational and behavioural change research. This method can be further extended by adding the collection and evaluation of location data to a study's protocol. However, there is a lack of a flexible and location-based software platform for researchers and clinicians to use the geographically explicit ecological momentary assessment approach. We filled this gap by developing the Healthy Environments and Active Living for Translational Health (HEALTH) Platform. The HEALTH Platform consists of a smartphone application for Android and iOS that can collect audio, photos, surveys, and videos; a logic server used to evaluate and respond to data from the app; and a monitor server for researchers and clinicians to review summaries of the collected data from the app. We demonstrate the HEALTH Platform's capabilities in a study of how exposure to parks may improve a person's well-being. The HEALTH Platform is an effective, flexible, and secure platform that can be applied in a wide range of digital health research and behaviour change contexts.

## 1. Introduction

The use of digital methods in health research has grown exponentially over the past two decades [1]. While dedicated sensors and personal wearables are a popular choice for technology-based health studies, smartphones have rapidly become the most common platform in digital health research [2,3]. The near ubiquitous access to cellular data-enabled smartphones also means participating in digital health research is now widely accessible in most developed countries [4].

Smartphones include a range of onboard sensors, and the functionality to connect wearable sensors, that can be used to measure physiological, locational, and contextual factors for health. For example, smartphone apps have been used to improve mental health [2,5,6], promote healthy dietary behaviours to tackle obesity [7–9], support people experiencing chronic disease [10–12], observe communicable disease patterns [13,14], and promote sustained change in health behaviours [3,15,16]. Smartphones are therefore an ideal delivery mechanism for observational and intervention-oriented studies of health behaviours and outcomes that support disease prevention and health promotion.

### 1.1 Ecological momentary assessment and pre-smartphone approaches

Ecological momentary assessment (EMA) is a methodological framework that encapsulates methods used to repeatedly measure how environmental and social contexts affect perceived or quantifiable health behaviours and/or outcomes over time [17,18]. These methods have also been referred to as ambulatory assessment, daily life

studies, experience sampling, or intensive longitudinal sampling [17,19,20]. In EMA protocols, participants are typically prompted on a random or fixed schedule to answer a short survey. Other EMA designs may involve participants independently responding to a survey whenever an event happens to them, such as craving a cigarette or making a purchase.

In pre-smartphone era study designs, participants would be asked to keep a diary or fill out a survey after being prompted through an electronic pager or watch-based alarm [20,21]. Yet these more manual methods are highly subject to recall bias where participants can complete the survey long after the occurrence of a relevant event. Moreover, these studies may be subject to desirability bias, where participants complete surveys all at once towards the end of a study to ensure they receive the full participation incentive available or performatively demonstrate to researchers' compliance with the protocol as a research participant [17,21].

The advent of digital EMA methods through smartphones has helped mitigate concerns related to recall and desirability bias in the data collection process [21–23], generating longitudinal data with greater validity. In comparison to non-digital EMA approaches, smartphones enable EMA prompts to be delivered and managed based on temporal context. Participants should respond to EMA prompts in the immediate context of a moment [18]. Such immediate EMA responses more effectively capture moment-to-moment experiences, thoughts, feelings, and/or behaviours, and the changes in these responses over time. Software controls can be used to limit the ability to complete prompts after a particular time period has elapsed, as well as measure the latency between a prompt and a response [17]. In short, smartphones have been a catalyst for a digital revolution in the many health and social science fields that make use of EMA methods.

## 1.2 Geographic ecological momentary assessment

Geographic ecological momentary assessment (GEMA) is an extension of the EMA method that incorporates the collection of background location data. The incorporation of geographic information enables data collection on environmental and contextual factors, in addition to the factors traditionally collected in EMA studies [24,25]. GEMA methods have been used to measure a wide range of health behaviours and outcomes including noise, diet, substance use, physical activity, and mental health [7,22,24,26–28]. The pioneering example of GEMA can be traced to the work of Swedish geographer Torsten Hägerstrand, who investigated the daily lives of people and their perspectives on place via paper-based diaries [29]. In the modern era of smartphones, sensors, and wearables, geographic data can now be collected frequently in time and precisely at a smaller geographic scale [30] or infrequently in time and coarsely at a broader geographic scale [31,32]. This geographic information can then be used to develop more sophisticated data linkages with other environmental data sources (based on location and time) to investigate how these location-based exposures may influence health behaviours and outcomes. Geographic data could also be used to trigger a survey prompt based on exposure to or engagement with a specific place like a park, commercial store, or transportation hub.

Most (digital) health interventions are theoretically informed by the social-ecological model of health, in which individual behaviours are nested within social relationships that are contained within an environmental and cultural context [33]. Health-related behaviours like diet, physical activity and social connection are affected by environmental exposures and engagement with others, which are compounded over time. The GEMA approach is an ideal way to record these momentary interactions across space and time and associate them with measures of health-related behaviours and health outcomes.

GEMA can involve the use of one or more approaches to delivering prompts to participants: time-contingent, event-contingent, location-contingent, and/or response-contingent. Time-contingent prompts are sent randomly or after a certain amount of time has elapsed between prompts. Time-contingent prompts are the most common form of prompt found in EMA studies; used to measure change over the course of a day or longer [17]. Event-contingent prompts involve prompting participants when an event occurs, measured through a sensor or having participants self-report information when a pre-determined event occurs to them. Measures of substance use, or stressful moments would typically make use of an event-contingent prompt design [34,35]. Location-contingent prompts involve the use of geofences or proximity

points to send prompts to participants when they are in a specific space and possibly at a specific time. For example, participants could be prompted when they are close to a grocery store [7] or within the boundaries of a greenspace [36]. Response-contingent prompts involve sending prompts based on past participant behaviours or responses. These prompts could involve tailoring survey questions, or altering the conditions required for sending other types of prompts to participants, based on previously recorded data from participants. A location-based prompt related to visiting a greenspace could then queue a later prompt to be sent about a participant's mental wellness.

GEMA can also resolve classical problems related to cross-sectional human research designs by enabling longitudinal research of space and time [37]. In addition, when GEMA is used to collect precise geographic information at a small enough scale, the geographic context can be better ascertained surrounding a participant at a singular point in time [38]. However, the collection of high-frequency geographic information absent of motivational context over a short time period means the information that is captured may not be an accurate representation of a participant's typical spatial behaviours [39]. Therefore, GEMA with survey prompts that record motivational reasoning, used across an extended time period, may capture more accurate spatial behaviour patterns. In summary, these improvements from GEMA deliver more reliable and valid measures of health behaviours and outcomes, resulting in more precise interventions for population health.

## 1.3 Motivation

Past GEMA studies have encountered challenges related to consistency across smartphone application platforms including the use of complex geometry for geofences, reliance on custom software configurations, and limited data collection and prompt types [24,25]. GEMA studies often require the development of bespoke software solutions that need constant updates to maintain compatibility with iOS (Apple) and Android (Google) operating systems [24,40]. This is a considerable problem for app-based research projects as compatibility issues frequently result in the exclusion of participants based on their smartphone's operating system or the need for different data collection protocols based on the type of operating system [40,41]. Software tools often used in previous GEMA studies include, but have not been limited to, Ethica Data, Happyhier, Itinerum, MASS, SmartAPPetite, and VERITAS [24,25,39,41]. Other studies have leveraged bespoke data collection tools that have extended existing survey tools, like Qualtrics or REDCap, to the maximum extent of their capabilities. These GEMA software platforms usually lack the ability to make use of complex dynamic geofences, instead relying on static proximity-based buffers to send location-contingent prompts [7,24]. Furthermore, these tools are limited to the collection of numerical or text-based survey responses [24,25], and the integration of sensor-collected data with smartphone-collected data only during analysis, instead of part of the study's data collection protocol [42].

To address these challenges, we sought to develop a digital health research platform for observational and intervention studies that would use the GEMA approach, while addressing key technical limitations faced in digital health research. A flexible and secure software platform could deliver better insights into the social and ecological contexts surrounding different health behaviours, as well as deliver personalized digital health interventions that could improve public health. The objective of this manuscript is to describe the design and functionality of such a software platform, illustrate its potential capabilities through a study on park exposure and well-being, and summarize how it addresses contemporary challenges associated with using the GEMA approach for digital health research.

## 2. Materials and methods

The Healthy Environments and Active Living for Translational Health (HEALTH) Platform is a flexible and secure software system for GEMA-based observational and intervention research studies. The HEALTH Platform is composed of three software components: (1) the HEALTH Platform App for Android and iOS operating systems, (2) the logic server used to evaluate and respond to data from the HEALTH Platform App, and (3) the monitor server for HEALTH Platform users, researchers, or practitioners to review data (Fig 1). The HEALTH Platform is intended to provide a digital option for health and social science researchers and/or practitioners to implement any form of (G)EMA-based observational or intervention study protocol.

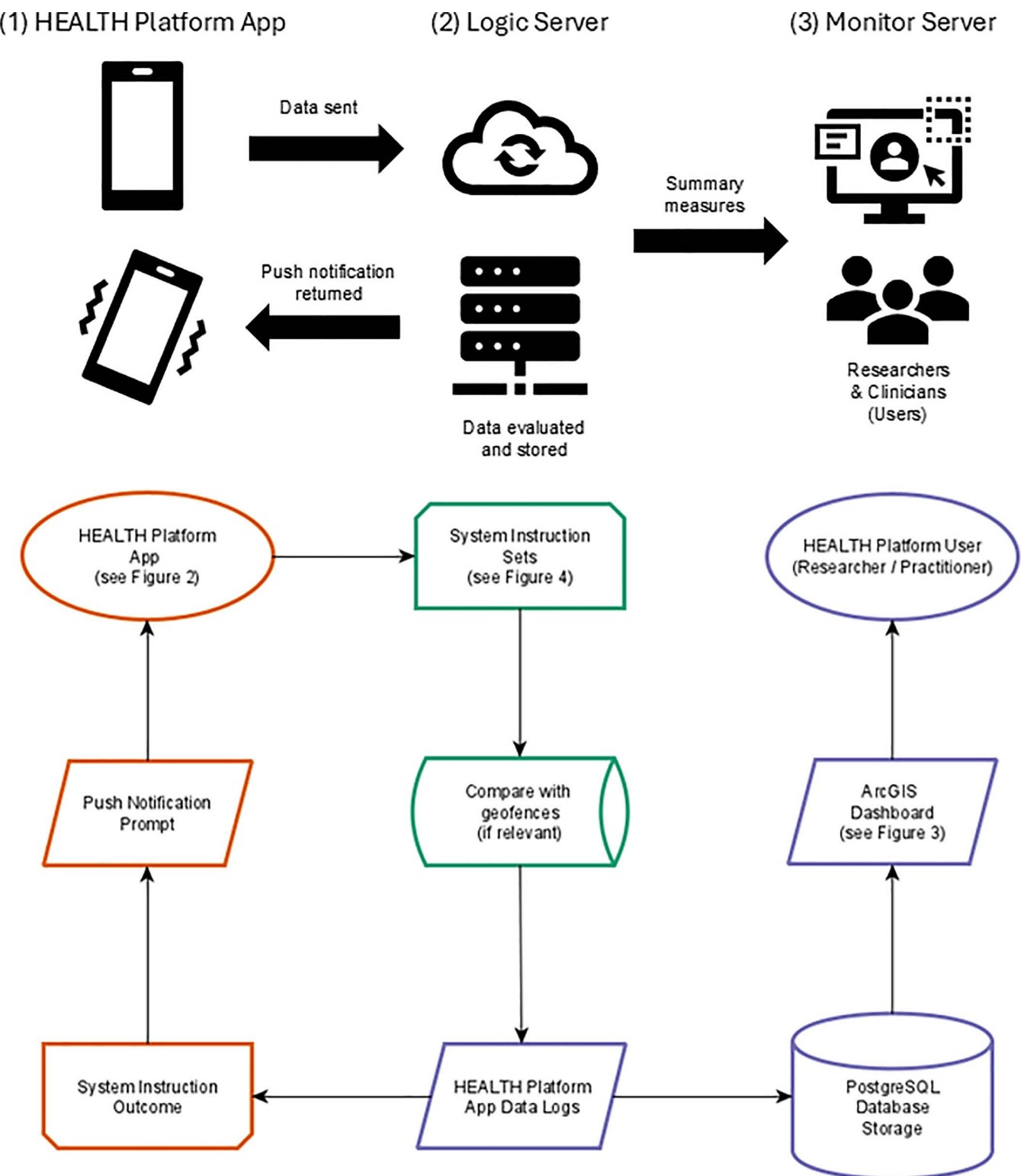

**Fig 1. A HEALTH Platform system architecture and conceptual data path between (1) HEALTH Platform App, (2) logic server, and (3) monitor server.**

The HEALTH Platform's general workflow for the setup of a study consists of evaluating the data types to be collected by the HEALTH Platform app, designing the system instruction sets that will be applied to that data on the logic server, and the layout of the reporting tools provided in the monitor server. As further described in the following sections, the HEALTH Platform app is designed to be used with any web or smartphone-based application to provide researchers with the greatest flexibility in using their preferred data collection method or tool. The logic server makes use of a low-code development environment to configure the system instruction sets that are applied to collected data. The monitor server makes use of a similar low-code development environment to provide additional flexibility to researchers and practitioners in designing and accessing reports about collected data. The HEALTH Platform is intended to provide an accessible and scalable system, leveraging open-source and freely available commercial software (for academic institutions), to undertake (G)EMA studies that do not rely on continual software development, ongoing system maintenance, or expensive subscription fees.

The HEALTH Platform's logic server can be configured to support prompts or intervention messages that are time-contingent, event-contingent, location-contingent, and/or response-contingent. These prompts could include traditional questionnaire-style requests, as well as more novel audiovisual requests that can be used to collect audio, photos, and/or videos. The platform is also capable of sending other textual, audio, video, or navigation-based messages to participants within the smartphone app. While the HEALTH Platform is intended to be used with a smartphone via the app to enable precise location-based data collection, it can also send and receive emails, text messages, phone calls, or instant messages to apps with an accessible application programming interface. This broadens the potential applications of the HEALTH Platform to other research designs that may wish to prompt and communicate with participants via methods other than app-based push notifications.

The HEALTH Platform app transmits, every two minutes, precise global navigation satellite system (GNSS) data containing the latitude and longitude coordinates of the device's position, the Unix Epoch time of the device, and a composite accuracy measure that represents the radial accuracy of the latitude-longitude coordinate pair's position in metres. GNSS locations are only collected while the device is in motion to conserve device battery life, with location data collection paused if the device remains motionless for longer than 10 minutes based on a combination of onboard accelerometry and positioning sensors. Location data collection resumes upon detection of movement based on the same sensors. This choice balances the collection of more detailed location information on individual movement with battery drain on user smartphones. All other types of events (e.g., push notifications and participant responses) are collected alongside GNSS coordinates. The platform automatically filters out location points that exceed 30 metres of radial inaccuracy.

Once a study or intervention protocol is configured, participants are provided with an invite code to access the platform and begin transmitting data. These invite codes allow the HEALTH Platform to support multiple studies with different protocols at the same time, as well as split participants between multiple control and intervention arms in the same study. The logic server receives data from the app, applies configured system instructions, and then acts based on the configured study or intervention protocol. This element of the platform can also consider previously collected data from the monitor server as part of the system instruction process. Survey administrators can then access a secure dashboard to review collected data and run preliminary analyses via the monitor server.

## 2.1 Ethics statement

The HEALTH Platform was reviewed and approved by The University of Western Ontario's Non-Medical Research Ethics Board (REB #121643) and the institution's Technology Risk Assessment Committee in January 2023. The smartphone app was reviewed and approved by the Apple App Store and Google Play Store for public distribution in June 2023.

## 2.2 Platform architecture

The HEALTH Platform's architecture stack consists of commercial software from Esri (esri.com, Redlands, CA, USA), the open-source PostgreSQL database software with the PostGIS extension, and React code written by authors AW, SZ, and

ML within the Expo Application Service ecosystem (expo.dev, Palo Alto, CA, USA). The logic server is an implementation of the Esri ArcGIS GeoEvent Server which makes use of a Java-based object-oriented programming language. Study or intervention protocols are translated into system instruction sets that render the desired actions to be taken in response to the received data. These system instruction sets are created by assembling a sequence of input connectors, filters, processors, and output connectors, as shown in the following section.

A smartphone app, listed on the Apple App Store and Google Play Store, was developed using Expo's React-based code library (https://expo.dev). The use of this codebase enables universal app experiences across all platforms and maintains app compatibility as operating system updates are made to iOS and Android. In addition, Expo's platform provides a service to deliver push notifications from the logic server to the app by interfacing seamlessly with Firebase Cloud Messaging (Android) and Apple Push Notification Service (iOS). The app can only be accessed by inputting an invite code provided by the study team. Prompts received via push notifications are completed by participants within the HEALTH Platform App using ArcGIS Survey123. However, the app can also be configured to use any form of survey software (e.g., Qualtrics, REDCap, Typeform) or link out to another application (including web browsers) present on the participant's device. Participants have the option on the home screen of the app to disable location logging via a toggle button. They are informed that they will no longer receive prompts to complete surveys should the location logging toggle be placed in the off position. Participants can also access summary measures about how many location points have been collected and how many prompts they have responded to during the study. Figures showing screen captures of the HEALTH Platform app and dashboards were removed from the manuscript due to this journal's copyright limitations.

The HEALTH Platform app itself is designed to be lightweight and highly efficient, ensuring compatibility with a wide range of mobile devices while minimizing demands on user hardware. The app only transmits location data and processes instructions received from the logic server. Prompts or messages sent from the logic server are handled by the app via a web browser or can be routed to another app installed on the participant's device. These system architecture design choices mean changes can be made rapidly to the HEALTH Platform's functionality without the need to push a software update to the app on a participant's device. It also means changes can be made rapidly to system instruction sets, how participants experience a study, allocations of participants to multiple complex treatment conditions within the same study or intervention protocol, and the design of dashboards and analysis tools available through the monitor server.

The HEALTH Platform smartphone app is connected to the logic server via Esri's ArcGIS REST application programming interface, with requests sent in GeoJSON format. Once data is received and processed by the logic server, it is transmitted to the monitor server. The monitor server stores the collected data in a PostgreSQL database, which can then be accessed and analyzed via the Portal for ArcGIS interface. Users who access the monitor server can review and analyze data collected using a highly configurable viewer built from ArcGIS Dashboard. If the configured system instructions require the evaluation of previously collected data, the logic server can securely access the data stored on the monitor server.

## 2.3 Data security

The HEALTH Platform is architected with a 'zero-trust' philosophy [43]. Identifying participant information (e.g., username, password, email) is not stored on the participant's device or the platform's logic and monitor servers. Instead, participants are issued an invite code by the study team to uniquely identify themselves within the data collected by the HEALTH Platform. This invite code is then used later offline to link HEALTH Platform data to a participant's other survey data using highly segmented database tables. Moreover, to resolve the concerns about the privacy of even anonymized geographic data logs collected from participants [44], the coordinates are disarranged on reaching the monitor server's storage using an algorithm to effectively mask the true coordinates of the data. The location data is then returned via the same algorithm to the correct coordinate positions during offline analysis only.

The platform's server hardware is located entirely on-premises in dedicated physical machines within institutional data centres. These physical machines are clustered to be locationally redundant with continual backups between them while also providing rapid failover capability. The logic and monitor server are located within a private virtual network that can only be publicly accessed by participants and users through a separate web server and load balancer. All data transmission is encrypted between the logic server, monitor server, and app on a participant's smartphone. The logic and monitor servers can only be accessed using HEALTH Platform-specific provisioned user accounts with two-factor authentication. Users of the dashboards on the monitor server can only access data associated with their study, enforced with user account group controls on the HEALTH Platform monitor server.

The HEALTH Platform's architecture and participant data controls conform to the privacy and security standards found in major national legal frameworks. These include the United States of America's *Health Insurance Portability and Accountability Act* Privacy Rule and Security Rule, the European Union's *General Data Protection Regulation* seven protection and accountability principles, and Canada's *Freedom of Information and Protection of Privacy Act* (Ontario) Collection Limitation principle and Data Security standards. The HEALTH Platform implements these frameworks by only collecting and storing data that is absolutely necessary for the operation of the platform, centralizing the platform to use only on-premises infrastructure exclusively controlled by the researchers, seeking informed and ongoing consent from all participants through the use of the invite codes, and providing participants with detailed controls over the data collection mechanisms and the ability to withdraw their data from the platform.

## 2.4 Platform functions

The HEALTH Platform's logic and monitor servers are highly customizable to enable a broad range of protocol designs. As stated previously, the logic server continuously evaluates data received from the app, as well as from the monitor server, applying the preconfigured system instructions to determine appropriate actions. Examples of these system instruction sets are provided in the following section. Any functionality available within Esri's ArcGIS Enterprise software ecosystem, or another tool with an accessible and secure application programming interface, can be configured within the HEALTH Platform's monitor server.

The logic and monitor servers assign a globally unique identifier to each record sent and received within the HEALTH Platform to enable data linkage. In the following sample configuration, responses to prompts within the HEALTH Platform App are completed using ArcGIS Survey123. When using ArcGIS Survey123, survey prompts can collect all types of traditional question designs while also providing the option for participants to submit up to 20 photographs, 20 minutes of audio, or 5 minutes of 720p video. The HEALTH Platform can also be configured to use other common survey platforms such as REDCap or Qualtrics.

Survey responses are received and stored by the monitor server, which can then be further evaluated by the logic server if desired in the study protocol. The app location data and prompt response data, once linked via tools available in the monitor server, can be aggregated into a wide range of spatiotemporal measures within the geographic information system present on the logic and monitor servers. It can also be compared with other geographic data to estimate the situational context of a location coordinate pair such as the land use type, presence within a geofenced area such as a park or retail area, or proximity to another geographic feature such as a store or pollutant source.

## 3. Results

The purpose of the following case study is to illustrate the potential of the HEALTH Platform for GEMA research. This example is not intended to provide any actionable and meaningful findings relevant to the hypothesized relationship between park exposure and health. Instead, the following sections provide methodological findings that exemplify the HEALTH Platform's improvements upon past (G)EMA methods. Moreover, the system instruction sets presented in this study demonstrate how the logic server can be configured to respond to collected data with location-contingent or time-contingent prompts.

Prior GEMA-based studies have attempted to establish the dose-response relationship between parks and well-being [22,32,36,45–47]. However, these studies had considerable spatiotemporal mismatch in which the outcome measure was collected outside of the area of exposure well after the exposure occurred in time. Further, many studies only captured a narrow time period lasting less than a week, and are therefore affected by the selective daily mobility problem [39]. Notably, a prior GEMA study of how natural environments affect happiness used location-contingent prompts that were delivered as participants experienced greener areas [36]. They also allowed participants to initiate their own survey responses using an event-contingent prompt design.

The HEALTH Platform's capabilities and potential applications are illustrated via a study that assessed the correlation between exposure to parks and a person's well-being and sleep quality. This illustrative study ("ParkSeek") consisted of recruiting a general population sample in Canada to participate in a similar study of how parks affect their well-being for approximately 14 days. For the ParkSeek study, participants had to reside in Canada, be at least 18 years of age, be able to comprehend English, and have access to a smartphone running iOS or Android with a consistent cellular data connection. There were no exclusion criteria. Consent was obtained in two stages from all participants. First, participants indicated their consent to the survey-based data collection at the onset of the study. Participants were then asked to consent again to the app-based component of the study which included explicit information about the precise location logging function of the HEALTH Platform app.

### 3.1 Study protocol: ParkSeek study on greenspace and well-being

ParkSeek study participants completed an entrance survey about their behaviours related to parks, perceived health and well-being, as well as sociodemographic characteristics like access to a vehicle, age, employment status, gender, income, and pet ownership. Participants were asked to install the HEALTH Platform app via the Apple App Store or Google Play Store on their own smartphone. They were issued an invite code to access the app's functionality, which once entered would begin transmitting data from their device to the HEALTH Platform's logic and monitor server. Participants were then asked to use the app for approximately 14 days.

Participants were asked to complete after the app-based study period elapsed an exit survey about their attitudes, behaviours, and perceptions towards parks and perceived health and well-being. These health and well-being measures were taken to provide a pre-study and post-study measure for each participant that could be used to account for common observational study biases, as well as incorporate person-level characteristics in subsequent statistical modelling of these data. Participants did not receive any specific training about how to use the app beyond the instructions presented on the home screen or as part of the entrance and exit survey instructions.

During the app-based data collection period, participants could receive four possible prompts sent via push notification: two different time-contingent-based prompts, and two different location-contingent-based prompts (Fig 2). Participants could be prompted to complete a survey response once between 9:00–10:00h local time (Fig 2A) and once in the evening between 19:00–20:00h local time (Fig 2B). Participants were only sent these prompts if their device transmitted a location within the relevant hour-long timeslot. Next, participants could be prompted to complete a survey whenever their device was detected to have been inside the geographic boundary of a park for at least 10 minutes (Fig 2C). If the participant's device remained inside the same park geofence for longer than 45 minutes, they would be prompted to complete the survey again as a follow-up measure. This time period was selected based on the associations between park exposure and health observed in previous studies [36,48,49]. Finally, participants were prompted to complete a momentary survey whenever their device exited the geographic boundary of a park (Fig 2D). All time-contingent and location-contingent prompts sent to participants expired one hour after the device successfully received the notification to reduce spatiotemporal mismatches between the exposure event and the survey response.

Participants were asked every morning to complete a survey about their perceived well-being in the past hour, using the World Health Organization-Five Well-Being Index (WHO-5), and their perceived sleep quality from the prior night

Legend

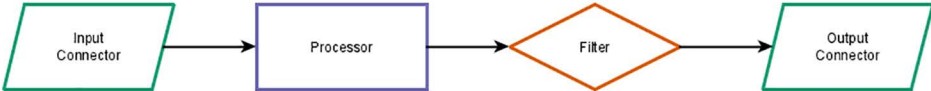

Panel A. Morning survey prompt

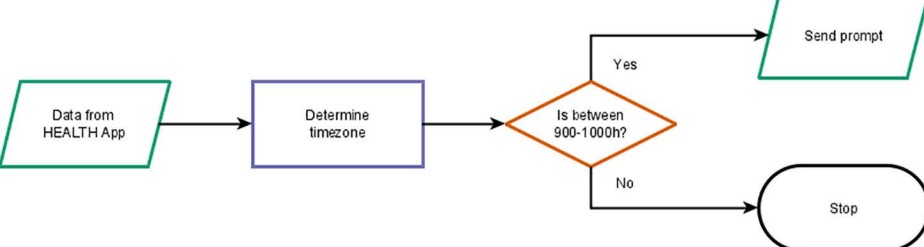

Panel B. Evening survey prompt

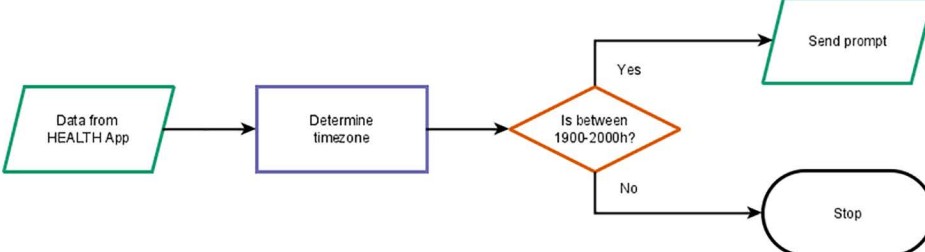

Panel C. Park engagement survey prompt

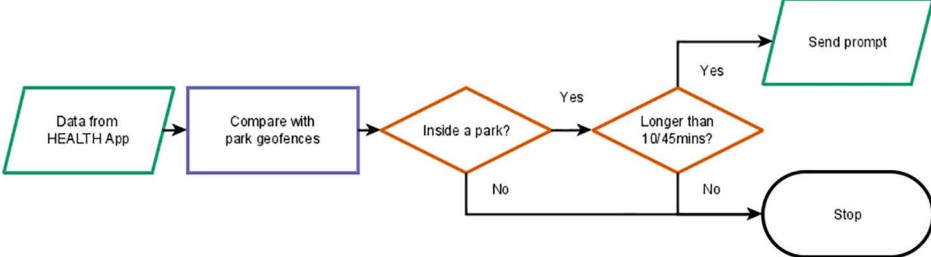

Panel D. Park exit survey prompt

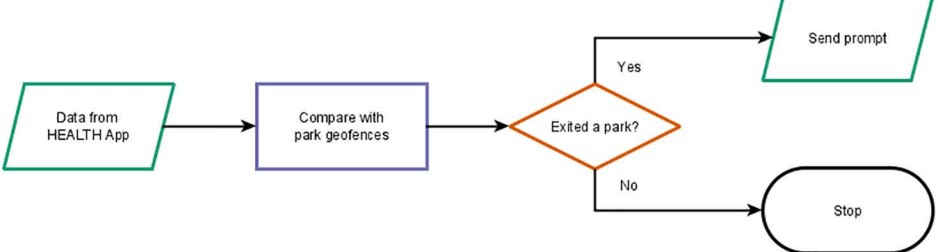

**Fig 2. System instruction sets used by the HEALTH Platform logic server to evaluate participant data.** Panel A shows the morning survey prompt instruction set. Panel B shows the evening survey prompt instruction set. Panel C shows the park engagement survey prompt instruction set. Panel D shows the park exit survey prompt instruction set.

using the Patient Report Outcomes Measurement Information System (PROMIS) Sleep Disturbance Index. The WHO-5 Well-Being Index questionnaire consists of five questions that sum to a total score of 25, with higher scores representing better well-being [50]. The PROMIS Sleep Disturbance Index questionnaire consists of 8 questions that sum to a total score out of 40, with lower scores representing better sleep quality [51]. Whenever participants were inside a park, they were prompted to record a short video about their experience and complete the WHO-5 Well-Being Index questionnaire. The recording of this video enabled the capture of the motivational context related to a park visit. Whenever participants exited a park, they were prompted to complete the WHO-5 Well-Being Index questionnaire. Participants were also prompted to complete the WHO-5 Well-Being Index questionnaire at the end of each day.

### 3.2 Preliminary results

Email invitations were sent to 1,379 participants who previously consented to be contacted about the ParkSeek study. Of those invitations, 96 consented to participate in the app-based ParkSeek study. The ParkSeek study undertook data collection from August to October 2023 using a continuous rolling recruitment model. Of the 96 that initially consented to participate in the study, 55 proceeded to download and access the app with their invite code. Among those 55 participants, 10 participants did not complete a single momentary survey prompt, and 7 had less than 7 unique non-continuous days of location data collected by the app. Therefore, 38 participants used the app by transmitting at least 7 unique days' worth of locations from their device, completing at least one momentary survey prompt in the app, and completing the exit survey. No issues were encountered with the delivery success of the prompts.

Participants had a median age of 40 years old. 60% of participants identified as women, and 15% identified as non-White. There were no gender, age, or other demographic differences identified between participants who had an acceptable level of compliance with the protocol as compared to those who did not.

A total of 330,294 unique locations were collected from these 38 participants during the study period. The average number of locations collected per participant was 8,692. These participants were prompted to complete a momentary survey in the app 1,373 times during the study period, with varying rates of adherence and latency in response times by prompt type (Table 1). In addition, participants submitted 17 videos as part of responses to the location-contingent park engagement prompts. Videos were submitted for 23% of the engagement-type prompts.

Participants displayed a diversity of geographic behaviours over the course of the study period. Some engaged more in recreation-focused parks while others spent considerable amounts of their time in nature-based parks. However, the vast majority of participants' time was spent in residential areas (>78% on average). Excluding the time spent in residential areas, participants spent 22% (SD = 7%) of their time in parks on average compared to all other destinations.

## 4. Discussion

The HEALTH Platform improves upon other GEMA tools by providing a flexible system architecture that supports a wide range of data collection methods, the means to vary prompts using complex location and/or time-based criteria, and the

**Table 1. Participant adherence and response latency by prompt type in the ParkSeek study.**

| Prompt Type | # Prompts sent | # Participants sent a prompt | % Adherence (St. Dev.) | Median latency in minutes (St. Dev) |
| --- | --- | --- | --- | --- |
| Daily Morning | 359 | 38 | 61% (30%) | 3 (13) |
| Daily Evening | 366 | 36 | 68% (28%) | 4 (14) |
| Park Engagement | 127 | 25 | 47% (41%) | 2 (12) |
| Park Engagement Follow-up | 29 | 11 | 50% (42%) | 9 (9) |
| Park Post-Engagement | 492 | 35 | 44% (34%) | 4 (10) |

capability to adjust system instruction sets at an individual participant level or in response to participant behaviour. The logic server can use complex geometries as part of the geofencing evaluation process in the system instruction sets, as compared to more traditional proximity or buffer-based approaches to the triggering of location-contingent prompts in other GEMA studies. In addition, the logic server can relate individual geographic features and their attributes to specific prompts or recorded data, instead of relying on post-data collection analyses, substantially increasing the nuance of logic applied in the system instruction sets, and spatiotemporal accuracy of collected data. In short, the HEALTH Platform is a transformative improvement for the conduct of GEMA studies.

## 4.1 Adherence to the ParkSeek study protocol

Population health research, especially when involving the general population with few inclusion or exclusion criteria, has traditionally struggled to achieve high levels of adherence to study protocols [52–54]. The ParkSeek study involved recruiting a general population from Canada with few inclusion or exclusion criteria. The invitation to participate in the ParkSeek study, sent only to individuals who had already indicated their willingness to participate in future studies, had a response rate of 7%. Of those 96 responding individuals, only 40% of participants adhered sufficiently to the study protocol. EMA as an intensive longitudinal observational method is primarily affected by adherence to prompted assessments, frequency of assessments, and the length of assessments. Other EMA studies, that report sufficient information to estimate adherence, are on average 79%, with the majority falling between 66–92% [18]. Similar adherence ranges have been reported in a systematic review of only GEMA studies [24]. Notably, the frequency of assessments has little effect on adherence while the length of assessments significantly affects adherence and dropout. The provision of incentives also does not affect participant adherence or dropout.

The adherence rates for momentary surveys in the ParkSeek study revealed a consistent trend that surveys with the lowest number of response items had the highest adherence rates, while the most complex or longest questionnaires had the lowest adherence rates. Further simplification of these questionnaires in future studies would likely improve adherence rates among a general population sample. The lower adherence observed in the ParkSeek study could also be related to participants being asked to complete surveys within parks, areas where technological separation is often desired to improve mental health [30]. Perhaps study designs that investigate other geographic contexts or health conditions would have greater adherence and participation rates.

## 4.2 Addressing selective daily mobility bias in the ParkSeek study

The collection of high-frequency location data is a potential boon to digital health research and practice. However, the collection of location data over a narrow timespan could encounter the selective daily mobility bias problem [39]. Essentially, only measuring a few days or a couple of weeks of a person's location behaviour may not accurately represent their typical behaviour. Extending the duration of the study or randomly sampling days over a longer time frame could be a potential solution to better capture typical behaviours. The ParkSeek study elected to collect data for 14 days, informed by previous work that identified this time period as sufficient for assessing complete activity spaces [55]. However, logging locations even over a longer time period without motivational context risks inferring erroneously that a particular behaviour took place somewhere. For example, if a participant was observed visiting a recreation centre, it would be equally reasonable to infer that they engaged in physical activity or were watching someone else (e.g., a child) engage in physical activity at that location. These inferences would likely cause Type I statistical errors in later modelling of how place affects health. The collection of video-based momentary surveys, as done in the ParkSeek study, provides the necessary information to address these potential errors.

While there was a low adherence to the video-related momentary survey in the ParkSeek study, participants received no guidance on recording videos beyond the instructions contained within the prompt. They also may have felt uncomfortable making these recordings in public settings like a park. A potential solution to this challenge would be to incorporate

lessons from the photovoice tradition [56]. These changes could include providing participants instruction in videography, building confidence in using the video-based method in a group setting, and collaboratively addressing concerns that participants may hold about the privacy and legality of making recordings in public spaces.

### 4.3 Lessons learned and potential applications of the HEALTH Platform

The HEALTH Platform is a flexible and secure tool for GEMA-based observational research and digital health interventions. The platform could be used to observe and intervene in a wide range of health behaviours and outcomes. More precise estimates can be made of the dose-response relationship between environmental exposures and health outcomes. For example, the HEALTH Platform could be used to measure how hearing aids perform in different acoustical environments [57], observe changes in perceived pain levels over time [58], or evaluate the impacts on mental health when commuting on public transit [26]. In an intervention context, the HEALTH Platform could be used to develop and monitor the uptake of a physical activity-based or social connectedness-oriented park prescription, whilst also measuring changes in perceptual quality of life measures [59]. It could be used to proactively inform people about the health risks associated with air pollution or sun exposure in their immediate environments [60]. In summary, there is considerable potential for the HEALTH Platform to make GEMA an accessible methodological option for digital health research and practice that requires inclusion of location-based measures in the intervention and/or analysis.

Participation in health behaviour interventions or intensive observational studies are often dependent upon population interest in the benefits to be gained from participation. For example, uptake is considerably low among all types of general population-targeted health interventions, except for those associated with HIV prevention [54]. In that case, there was a direct tangible outcome from research participation: not being infected with HIV. Clinical populations show similar patterns with those most affected by chronic disease, and therefore having higher health literacy, showing higher engagement with observational and intervention studies than the general population [61]. The HEALTH Platform may experience greater uptake and adherence among more clinical populations that are motivated to participate because there are tangible benefits to realize for their health.

A common proposal to improve general population uptake is the incorporation of theoretical behaviour change models [62] and personalized interventions [63]. The HEALTH Platform, as a digital health tool, presents an ideal method to enact these proposals, as the system could be extended to incorporate algorithmic responses that dynamically adjust to participant behaviours and interests over time. For example, large language models could be used to create personalized activity coaching that is further supplemented by incorporating real-time geographic information and participant feedback on that coaching [64]. This approach could be easily integrated within the HEALTH Platform's logic and monitor servers.

Interventions could also be personalized to participants within the HEALTH Platform logic server based on their stated preferences and observed behaviour. Gamification, tailored incentives, and other behavioural economics-informed strategies could be leveraged to improve engagement with the app. For example, in the context of an intervention study like the ParkSeek study, the HEALTH Platform could hypothetically incorporate an algorithm that evaluates participant well-being changes after visiting a park, then recommend future parks to visit that best match the quality of conditions experienced in parks that were correlated with the greatest improvement in well-being for that individual participant. Participants could potentially earn points to exchange for free recreation classes held at local parks and recreational facilities, all managed through the HEALTH Platform's logic server and monitor server. In turn, health promotion professionals and participants could collaboratively identify improvements to the study's intervention, resulting in more effective practice.

The HEALTH Platform potentially addresses the selective daily mobility bias by collecting qualitative information from surveys and audiovisual recordings about the activities, motivations, and conditions experienced by participants in place. These videos can be used to avoid making possibly erroneous assumptions about the motivations for visiting a place, or the types of behaviours occurring in that place. In prior studies, videos have been identified as a rich source of supplemental information to contextualize location patterns and other health measures [22,27,65,66]. Continued improvement to

the HEALTH Platform's audiovisual data collection functionality and participant instructions could make valuable contributions to addressing the contemporary biases encountered in location-based digital health research.

### 4.4 Implications and contributions

The HEALTH Platform addresses multiple technical challenges that have been experienced in past GEMA studies [24,25]. The approach to software development provides flexibility to accommodate all potential research designs, while also using privacy-preserving techniques that protect participants. The use of the Expo Application Service ecosystem reduces the frequency of updates to the smartphone app, while also ensuring consistent experiences for participants across Android and iOS operating systems. The HEALTH Platform has a considerable advantage over existing GEMA software platforms, given the logic server can make use of complex geofence boundaries, as opposed to proximity points or circular buffers, when evaluating location data as part of the system instruction sets. Furthermore, the HEALTH Platform can collect a wide range of data types including numerical, text, audiovisual, and integrated sensor data from participants. In addition, the HEALTH Platform results in more personalized and participant-led research practices. Traditional mixed methods research has typically required the presence of a researcher with a participant to make observations and ask questions of the participant while they move through a place [67]. The use of an app to collect data in an audiovisual format provides participants with the agency to share their perspectives on place and health in their own words. Marginalized participants may feel more comfortable revealing their true thoughts about how they feel in a place in the absence of an accompanying researcher.

The lack of a flexible software platform for GEMA research led to the development of the HEALTH Platform. We illustrated the potential use of this platform through a study of how park exposure may affect well-being and sleep quality. The HEALTH Platform addresses many concerns that have been raised about privacy, adherence, technical complexity, and sampling challenges associated with GEMA research and practice. The ubiquity of smartphones means app-based protocols like the HEALTH Platform are feasible and accessible across multiple population groups. As compared to traditional ambulatory assessment methods, GEMA methodologies enable the collection of spatiotemporal data that can be used to develop highly detailed co-variates to be incorporated into an analysis. For example, geographic data logs could be used to estimate the amount of time spent engaged with specific amenities inside a park, estimate exposures to greenness outside of parks, or determine other amenities like fitness centres or grocery stores that were visited by participants.

The ParkSeek study illustrates the commonly found adherence problem in general population health research. On the other hand, the HEALTH Platform presents a novel solution to the selective daily mobility bias by collecting motivational context with video recordings made by participants. Future studies may wish to more comprehensively evaluate the performance of the HEALTH Platform as compared to other software platforms when used by the same group of participants. Implementation of the HEALTH Platform in a clinical trial or public health practice setting are also desirable areas for future studies. These studies could be used to identify and address barriers to implementation such as technological adoption, patient and provider data privacy concerns, and the scalability of the system to multiple treatment conditions. Overall, the implications for future research and policy from the HEALTH Platform consist of more inclusive research designs, improved precision in estimating dose-response relationships between environmental covariates and health behaviours, and improved targeting of health promotion interventions for population health policymakers.

## Author contributions

**Conceptualization:** Alexander James David Wray, Sean Doherty, Jason Gilliland.

**Data curation:** Alexander James David Wray, Shiran Zhong.

**Formal analysis:** Alexander James David Wray, Sean Doherty, Jed Long, Jason Gilliland.

**Funding acquisition:** Alexander James David Wray, Sean Doherty, Jed Long, Christopher J Lemieux, Jon Salter, Jason Gilliland.

**Investigation:** Alexander James David Wray, Katelyn R O'Bright, Shiran Zhong, Sean Doherty, Catherine E Reining, Christopher J Lemieux, Jason Gilliland.

**Methodology:** Alexander James David Wray, Katelyn R O'Bright, Shiran Zhong, Sean Doherty, Michael Luubert, Jed Long, Jason Gilliland.

**Project administration:** Alexander James David Wray, Catherine E Reining, Christopher J Lemieux, Jason Gilliland.

**Software:** Alexander James David Wray, Shiran Zhong, Michael Luubert.

**Supervision:** Sean Doherty, Christopher J Lemieux, Jason Gilliland.

**Visualization:** Alexander James David Wray, Sean Doherty, Jed Long.

**Writing – original draft:** Alexander James David Wray.

**Writing – review & editing:** Katelyn R O'Bright, Shiran Zhong, Sean Doherty, Michael Luubert, Jed Long, Catherine E Reining, Christopher J Lemieux, Jon Salter, Jason Gilliland.

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
