## [Decision Letter · Decision Letter 0]

21 Jul 2025

Response to Reviewers
Revised Manuscript with Track Changes
Manuscript
**Journal Requirements:**

1. Please ensure that your Ethics Statement is available in its entirety at the beginning of your Methods section, under a subheading 'Ethics Statement'.

2. Please upload separate figure files in .tif or .eps format. Also, remove the figures from your manuscript file but keep the legends.

3. For studies involving third-party data, we encourage authors to share any data specific to their analyses that they can legally distribute. PLOS recognizes, however, that authors may be using third-party data they do not have the rights to share. When third-party data cannot be publicly shared, authors must provide all information necessary for interested researchers to apply to gain access to the data. (https://journals.plos.org/plosone/s/data-availability#loc-acceptable-data-access-restrictions)

4. Some material included in your submission may be copyrighted. According to PLOS’s copyright policy, authors who use figures or other material (e.g., graphics, clipart, maps) from another author or copyright holder must demonstrate or obtain permission to publish this material under the Creative Commons Attribution 4.0 International (CC BY 4.0) License used by PLOS journals. Please closely review the details of PLOS’s copyright requirements here: PLOS Licenses and Copyright. If you need to request permissions from a copyright holder, you may use PLOS's Copyright Content Permission form.

Potential Copyright Issues:

Figure 2: contains branding/a logo. We are not permitted to publish this under our CC-BY 4.0 license, even with permission. We ask that you please remove or replace it.

Figures 2, 3, and 5: contains screenshots. We are not permitted to publish these under our CC-BY 4.0 license; websites are usually intellectual property and are copyrighted. This includes peripheral graphics of the web browser such as the [X] buttons. We ask that you please remove or replace it.

**Additional Editor Comments (if provided):**

However, some sections, particularly in the discussion, would benefit from tighter narrative cohesion, clearer alignment with referenced literature, and more critical reflection on the limitations of the ParkSeek study.  Additionally, the reviewers suggest elaborating on privacy concerns, usability issues, and adding comparison with existing platforms. Please see their comments in detail below.

**Reviewers' Comments:**

**Comments to the Author**

1. Does this manuscript meet PLOS Digital Health’s publication criteria?

Reviewer #1: Partly

Reviewer #2: Yes

Reviewer #3: Yes

2. Has the statistical analysis been performed appropriately and rigorously?

Reviewer #1: Yes

Reviewer #2: N/A

Reviewer #3: I don't know

3. Have the authors made all data underlying the findings in their manuscript fully available (please refer to the Data Availability Statement at the start of the manuscript PDF file)?

Reviewer #1: Yes

Reviewer #2: Yes

Reviewer #3: No

4. Is the manuscript presented in an intelligible fashion and written in standard English?

Reviewer #1: Yes

Reviewer #2: Yes

Reviewer #3: Yes

Reviewer #1: I generally found the HEALTH platform presented in this paper is innovative and technically superior and potentially could make it easier for the community to conduct GEMA studies. It addresses several challenges in current GEMA software platforms by having benefits including device-agnostics, update-free deployment and multiple data element support. The architecture appears to be robust, flexible and well explained. The ParkSeek Study is a solid study with interesting findings around survey prompt adherence. However, I did not find the presentation of the ParkSeek Study is able to fully demonstrate the flexibility of the mobile application and the capacity or advantage of meeting critical quality standard in GEMA field. Here are some comments.

1. The workflow for researchers:

The paper does not sufficiently explain what the researcher setup process looks like, which is important to demonstrate the flexibility and usability of the platform. How could the ParkSeek survey be easily deployed to the platform without customized development? How customizable is it in practice for people with different skills? How easy is it to add different data type to the survey? I would recommend the paper takes some effort explaining this process using the ParkSeek Study.

2. Clarify how the platform impacts the cast study:

The ParkSeek Study does not clearly articulate how the platform’s development contributed to solving specific challenges compared to previous tools or custom apps. What would have been difficult or impossible to achieve without the HEALTH platform? What specific problems faced by prior park exposure studies were overcome here? What kind of quality standard is met? A deeper analysis would strengthen the case that HEALTH is not only technically superior but practically transformative.

3. Distinguish conclusion, discussion and future work section

The current ending discussion sections include some discussion around best practice of GEMA studies, various existing challenges and how the HEALTH platform could potentially contribute to addressing those challenges, and general considerable advantages of HEALTH platform. But everything is blended. A more clear structure with the key findings of conducting ParkSurvey with the HEALTH platform, the current contribution how HEALTH platform is improving the GEMA studies, and areas for future improvement of HEALTH platform would significantly improve readability and uptake by readers.

4. Some other minor comments:

a. What kind of Privacy and Security standard the platform is sticked to? HIPAA/GDPR?

b. According to the paper, “Any functionality available within Esri’s ArcGIS Enterprise, software ecosystem can be configured within the HEALTH Platform’s monitor server.” What kind of configuration would it need, coding/pseudo-coding? Any limit of relying on Esri’s ArcGIS Enterprise?

Reviewer #2: Review for : The Healthy Environments and Active Living for Translational Health 2 ‎;‎;(HEALTH) Platform: A smartphone-based platform for geographic ecological 3 ‎;momentary assessment research

Title: These are my concerns:‎;

‎;-‎; Try to capitalize the first letter of each word of the title to make it coherent.‎;

‎;-‎; ‎; he word “platform” is used twice and should be revised to avoid repetition.‎;

‎;-‎; It combines many academic keywords (e.g., ecological momentary assessment, ‎;geographic, translational health), which may reduce clarity and immediate ‎;comprehension.‎;

‎;-‎; Shorten the title to focus on the scope and impact.‎;

‎;-‎; Examples: The HEALTH Platform: A Smartphone-Based Tool for Geographic ‎;Ecological Momentary Assessment or The HEALTH Platform for Translational ‎;Health: Smartphone-Based Geographic Ecological Momentary Assessment

The abstract, these are my comments:‎;

‎;-‎; The abstract contains long sentences and overuses technical phrases, making it ‎;harder for a general audience to absorb.‎;

‎;-‎; Results from the illustrative study are vague; no quantitative findings or sample ‎;characteristics are provided.‎;

‎;-‎; The statement “improvements upon existing GEMA platforms” is too general—‎;what specific enhancements were made (e.g., geofencing precision, data ‎;flexibility)?‎;

‎;-‎; Line 20–21: “Smartphones are an increasingly popular delivery mechanism...”

‎; Suggest: “Smartphones have become a widely used tool for delivering digital ‎;health interventions and conducting observational research.”‎;

‎;-‎; Line 22–23: “which can be extended by incorporating the collection of participant ‎;location data...”

Suggest: “which can be enhanced by collecting participant location data using ‎;built-in smartphone technologies.”‎;

‎;-‎; Line 24: “customizable, geographically appropriate software”

Suggest: “customizable software capable of supporting geographically explicit ‎;research.”‎;

‎;-‎; Line 29–30: “without the need for time-consuming updates...”

Suggest: “without requiring time-consuming updates to participants’ devices.”

‎;-‎; Line 31–32: “This study demonstrates the HEALTH Platform’s improvements ‎;upon existing GEMA software platforms.”

‎;→ Suggest: “This study illustrates how the HEALTH Platform improves upon ‎;existing GEMA software by offering greater customization and real-time ‎;flexibility.”‎;

‎;-‎; Line 34–35: “Overall, we find that the HEALTH Platform is a flexible mobile phone ‎;application...”

Suggest: “Overall, the HEALTH Platform offers a flexible solution for implementing ‎;GEMA in digital health research.”‎;

The introduction:‎;

• The section is quite long and could benefit from clearer sectioning or condensation ‎;of overlapping ideas (e.g., multiple discussions of smartphone EMA vs pre-digital ‎;EMA).‎;

• The research objective (lines 161–164) comes very late. It would be more effective ‎;to briefly state the study’s purpose at the beginning and then expand upon it later.‎;

• Some claims (e.g., on technical limitations of GEMA platforms) heavily rely on the ‎;same few sources ([24], [25], [40]), which weakens the perception of broad ‎;evidence.‎;

• While limitations of existing GEMA platforms are described, the authors could be ‎;more explicit about why addressing these limitations matters for scientific or ‎;population health outcomes.‎;

• Line 60: “has become almost universally accessible to the general population in ‎;developed nations.” Suggest: “is now widely accessible in most developed ‎;countries.”‎;

• Line 65: “observe communicable disease patterns [13,14, and promote sustained ‎;change…” Correction: Missing closing bracket: [13,14]‎;

• Line 84–85: “addressed many of these concerns about recall and desirability ‎;bias…” Suggest: “helped mitigate concerns related to recall and desirability ‎;bias…”‎;

• Line 87–88: “based on temporal context, that is participants respond…” Suggest: ‎;‎;“based on temporal context—that is, participants respond…”‎;

• Line 110–111: “theoretically informed by the social-ecological model of health, ‎;where…” Suggest: “…model of health, in which individual behaviors…”‎;

• Line 142–161 (Motivation paragraph): Overly dense. This could be split into ‎;two paragraphs: one detailing the problems with existing GEMA tools and a ‎;second outlining the authors’ solution and objectives.‎;

• Line 157: “…while also addressing the current technical limitations found in the ‎;digital health research area.” Suggest: “…while addressing key technical ‎;limitations currently faced in digital health research.”

Materials and Methods: ‎;

• Some sections (especially on architecture and software libraries) may overwhelm ‎;readers without computer science backgrounds. Consider moving these to an appendix ‎;or supplementary material.‎;

• The ParkSeek study lacks information on sample size, demographics, adherence rates, ‎;and outcomes, which are crucial for assessing feasibility and generalizability.‎;

• The authors should acknowledge technical or practical limitations (e.g., battery drain, ‎;GPS inaccuracies, reliance on Esri licenses).‎;

• No benchmarks or error rates are reported for prompt delivery success, location ‎;accuracy, or video data quality.‎;

• Only one pilot study is provided, with no comparison to other EMA/GEMA tools or ‎;platforms.‎;

• Repetitive phrasing: “The HEALTH Platform app…” and “logic server…” appear ‎;excessively. Use pronouns or merge sentences for variety.‎;

• Redundant language: “A flexible and secure software system…” and later “flexible ‎;unified software platform…” try to revise for variation and conciseness.‎;

• Improper punctuation and spacing: Line 65: “promote sustained change in health ‎;behaviours [3,15,16].” Ensure proper spacing and citation bracket closure (which was ‎;missing earlier).‎;

• Overuse of passive voice: E.g., “data is transmitted,” “survey is prompted,” “responses ‎;are recorded.” Consider active voice where clarity permits.:‎;

• Lines 243–249 and 295–307 could be broken into shorter sentences to improve ‎;readability.‎;

• Ensure consistent use of past tense when describing completed studies (e.g., ‎;ParkSeek), and present tense when referring to platform functionality.‎;

Results:‎;

• Only 38 participants met the inclusion threshold out of 1,379 contacted. This ‎;significantly limits the generalizability and statistical power of the results.‎;

• The section does not report findings on how park exposure influenced well-being or ‎;sleep scores, even descriptively.‎;

• Race, socioeconomic status, education level, or geographic region (urban/rural) are ‎;underreported despite being highly relevant.‎;

• No t-tests, regression models, or other statistical analyses were presented to assess ‎;adherence differences or behavioral outcomes.‎;

• While video submission rates are mentioned, no content analysis, coding, or ‎;qualitative findings are discussed.‎;

• ‎; Line 311:“whereby the outcome measure for participants was measured outside…” ‎;Suggest: “in which the outcome measure was collected outside…”‎;

• Line 314: “therefore being affected…” Suggest: “and were therefore affected…”‎;

• Lines 324–326: Redundant: “Consent was obtained in two stages... participants ‎;indicated their consent...” Suggest: Combine into one sentence for flow: “Consent ‎;was obtained in two stages, including explicit agreement to the location logging ‎;function.”‎;

• Lines 329–344: This paragraph could be split into two: one for the onboarding/survey ‎;process and one for post-study assessment.‎;

• Lines 380–392: The phrase “participants were also prompted to…” is repeated. ‎;Suggest: Vary the structure for better readability.‎;

• Table 1 title: Title should specify that the table includes both adherence and latency ‎;statistics, e.g., “Participant Adherence and Response Latency by Prompt Type in the ‎;ParkSeek Study”‎;

• Line 417: “...a video was submitted 23% of the time” Suggest: “Videos were ‎;submitted in 23% of engagement-type prompt responses.”‎;

Discussion:‎;

‎;-‎; The authors could expand on strategies for improving participant engagement, ‎;especially in low-adherence environments (e.g., parks). Try to include more ‎;concrete future recommendations (e.g., gamification, tailored incentives).‎;

‎;-‎; Low adherence to video responses undermines this proposed advantage. Suggestion: ‎;Propose detailed implementation of “photovoice” training or alternative non-intrusive ‎;context capture methods.‎;

‎;-‎; Add a sentence indicating what features are already operational and which are ‎;prospective.‎;

‎;-‎; Consider including a brief ethical discussion on audiovisual data usage, ‎;especially in public settings.‎;

‎;-‎; Line 434: “only 40% achieved an acceptable level of adherence to the study ‎;protocol.” Consider: “Only 40% of participants adhered sufficiently to the study ‎;protocol.”‎;

‎;-‎; Line 437: “adherence with responding to prompted assessments” Should be: ‎;‎;“adherence to prompted assessments”‎;

‎;-‎; Line 445: “most complex and/or longest questionnaires” Prefer: “most complex or ‎;longest questionnaires”‎;

‎;-‎; Line 454: “there is considerably low uptake” Correct to: “there is considerably low ‎;uptake” → “uptake is considerably low”‎;

‎;-‎; Line 463: “models of behaviour change and personalized interventions.” Parallel ‎;structure might benefit from: “behavior change models and personalized ‎;interventions”‎;

‎;-‎; Line 474: “may not be an accurate representation of their normal behaviours” ‎;suggests: “may not accurately represent their typical behavior”‎;

‎;-‎; Line 492: “participants were not given any guidance on how to record videos” ‎;suggests: “participants received no guidance on recording videos”‎;

‎;-‎; Line 517: “that needs to include” Improve to: “that require inclusion of”‎;

‎;-‎; Line 529: “ensures flexibility to accommodate all potential research designs” More ‎;precise, “provides flexibility to support a wide range of research designs”‎;

‎;-‎; Line 551: “highly accessible and feasible for use among a range of populations.”

Rephrasing suggest: “feasible and accessible across diverse populations.”‎;

References:‎;

• Reference numbers [e.g., 12, 61] include the journal name but not consistently in ‎;italics. Standard referencing style (e.g., AMA, APA, Vancouver) should be applied ‎;uniformly.‎;

• There are inconsistencies in journal abbreviations (e.g., JMIR mHealth uHealth vs. J ‎;Med Internet Res vs. PLOS Digit Health). Consider standardizing based on the target ‎;journal's guidelines.‎;

• Ref. 61 (O’Bright K) is a valuable addition but should be clearly marked as a thesis ‎;and cross-checked for DOI or permanent repository link format.‎;

• References 2, 5, 6, and 15 are all meta-reviews on mobile interventions—consider ‎;streamlining to reduce overlap unless each is cited for a distinct contribution.‎;

• References such as [17], [20], [21] (Stone, Mehl, Silvia) are essential and well-cited; ‎;however, ensure that the main text leverages their methodological frameworks ‎;explicitly.‎;

• Ref. 4: The citation of the Ericsson Mobility Report includes a URL that may need ‎;formatting as per the journal style (e.g., accessed date placement).‎;

• Ref. 29: Appears to list “Pred A” twice; double-check for typographical duplication in ‎;author names.‎;

• Ref. 56 and 57: The authors “Astell-Burt et al.” and “Barber et al.” appear in the same ‎;paragraph in the discussion—ensure the narrative distinguishes their contributions in-‎;text.‎;

Reviewer #3: This manuscript introduces the HEALTH Platform, a customizable smartphone-based system for implementing geographically explicit ecological momentary assessment (GEMA) in digital health research. The authors present the platform's design, architecture, and capabilities, and illustrate its functionality through the ParkSeek study, which investigates associations between park exposure and well-being.

The study is timely and addresses a clear gap in the digital health landscape, particularly in the availability of flexible, privacy-preserving, and scalable tools for GEMA research. The manuscript is well-structured and generally clear, although there are areas that would benefit from further clarification and discussion.

Pros:

- Platform fills a critical void in GEMA tools, offering a flexible, secure, and cross-platform solution that supports audiovisual data, dynamic geofencing, and real-time logic control.

- The authors present a unique contribution that goes beyond traditional EMA methods.

Cons:

- While the manuscript notes adherence challenges, the authors can discuss usability issues, including qualitative feedback from participants or a user study.

- More insight into why participants avoided video recordings and how this might be addressed (e.g., training, incentives, privacy messaging) would strengthen the discussion.

- How can the platform made more scalable? The authors can also discuss the potential barriers to adoption of this system in public health practice or clinical trials.

- Maybe elaborate on built-in data analysis pipelines or tools available to users through the monitor server.

- Comparison with Existing Tools: A concise table or section comparing the HEALTH Platform with other GEMA-capable apps (e.g., Itinerum, Beiwe, or bespoke REDCap-GIS integrations) would contextualize its novelty more clearly.

- The description of prompt delivery logic (especially around “park engagement follow-up”) can be overly technical in places. Consider simplifying or visualizing the logic to improve reader comprehension. This might increase the reproducibility of the model.

Minor comments:

- Check that the formatting of figures and citations are consistent. An external check is needed.

- Consider including example visualizations from the dashboard interface.

- Is the app is open-source or available to other researchers and under what licensing terms.

**Do you want your identity to be public for this peer review?** For information about this choice, including consent withdrawal, please see our Privacy Policy

Reviewer #1: No

Reviewer #2: **Yes: ** Prof. Ibrahim Shady (MBBChs, MPH, DrPH, MD, FRSPH, IMFPH, IMEUPHA)

Reviewer #3: No

**Figure resubmission:****Reproducibility:** To enhance the reproducibility of your results, we recommend that authors of applicable studies deposit laboratory protocols in protocols.io, where a protocol can be assigned its own identifier (DOI) such that it can be cited independently in the future. Additionally, PLOS ONE offers an option to publish peer-reviewed clinical study protocols. Read more information on sharing protocols at https://plos.org/protocols?utm_medium=editorial-email&utm_source=authorletters&utm_campaign=protocols

---

## [Decision Letter · Decision Letter 1]

24 Nov 2025

The Healthy Environments and Active Living for Translational Health (HEALTH) Platform: A smartphone-based system for geographic ecological momentary assessment research

PDIG-D-25-00222R1

Dear Dr. Wray,

We are pleased to inform you that your manuscript 'The Healthy Environments and Active Living for Translational Health (HEALTH) Platform: A smartphone-based system for geographic ecological momentary assessment research' has been provisionally accepted for publication in PLOS Digital Health.

Best regards,

Aline Lutz de Araujo

Section Editor

PLOS Digital Health

**Additional Editor Comments (if provided):**

**Reviewer Comments (if any, and for reference):**

Reviewer's Responses to Questions

**Comments to the Author**

Reviewer #1: All comments have been addressed

Reviewer #2: (No Response)

publication criteria?

Reviewer #1: Yes

Reviewer #2: Yes

3. Has the statistical analysis been performed appropriately and rigorously?

Reviewer #1: N/A

Reviewer #2: Yes

4. Have the authors made all data underlying the findings in their manuscript fully available (please refer to the Data Availability Statement at the start of the manuscript PDF file)?

Reviewer #1: Yes

Reviewer #2: Yes

5. Is the manuscript presented in an intelligible fashion and written in standard English?

Reviewer #1: Yes

Reviewer #2: Yes

Reviewer #1: (No Response)

Reviewer #2: Substantive improvements were implemented across the abstract, phrasing in the introduction, many discussion edits (engagement strategies, photovoice, wording fixes), the table title, and several style cleanups.

Still to do (high-impact):

Title: choose a shorter, cleaner version (one of your suggested options).

Methods: move architecture/vendor detail to supplement, and add a clear limitations paragraph (battery, GPS accuracy/drift, geofence miss/false-positive risk, dependence on Esri licenses, device heterogeneity).

Benchmarks: add prompt delivery success rate, location accuracy metrics (e.g., median HDOP/accuracy bucket), and any video capture success/quality stats.

Results: at least add descriptive associations (or explicitly flag as out-of-scope in the abstract and title as a platform/methods paper), and broaden demographics (SES/education/urbanicity) if available.

References: normalise to the target journal’s style and trim overlapping meta-reviews.

**Do you want your identity to be public for this peer review?** For information about this choice, including consent withdrawal, please see our Privacy Policy

Reviewer #1: No

Reviewer #2: **Yes: ** Ibrahim Shady
